# *Cryobacterium Inferilacus* sp. nov., a Pshychrophilic Ureolitic Bacterium From Lake Untersee in Antarctica

**DOI:** 10.3390/microorganisms13050990

**Published:** 2025-04-25

**Authors:** Yulia Yu. Berestovskaya, Tatyana P. Tourova, Denis S. Grouzdev, Natalyia V. Potekhina, Dmitry S. Kopitsyn, Nikolay V. Pimenov, Lina V. Vasilyeva

**Affiliations:** 1Winogradsky Institute of Microbiology, Research Center of Biotechnology, Russian Academy of Sciences, Moscow 119071, Russia; tptour@rambler.ru (T.P.T.); npimenov@mail.ru (N.V.P.);; 2SciBear OU, 10115 Tallinn, Estonia; denisgrouzdev@gmail.com; 3Faculty of Biology, Lomonosov Moscow State University, Moscow 119234, Russia; potekhina56@mail.ru; 4Department of Physical and Colloid Chemistry, Faculty of Chemical and Environmental Engineering, Gubkin University, 65-1, Moscow 119991, Russia; kopicin.d@inbox.ru

**Keywords:** psychrophilic bacterium, low-temperature ecosystem, *Cryobacterium*, family *Microbacteriaceae*, Antarctica, actinomycetes, genome

## Abstract

The psychrophilic aerobic heterotrophic bacterium, strain 1639^T^, was isolated from the low-temperature Lake Untersee in Antarctica. The bacterium was Gram-positive, non-motile, yellow–green-pigmented, non-spore-forming, and a pleomorphic rod. Growth was observed at temperatures of 0–25 °C with an optimum at 10 °C. The strain used urea as a nitrogen source. The major fatty acids were i-C_16:0_ (49.69%), ai-C_15:0_ (17.59%), and C_16:1_ branched (12.03%). Identified polar lipids were phosphatidylglycerols and a glycolipid. The respiratory quinone was determined to be MK-10. The genomic DNA G+C content was 68.03 mol%. Phylogenetic analysis based on 16S rRNA gene sequences indicated that strain 1639^T^ was a member of the genus *Cryobacterium*, with the highest sequence similarity to *C. arcticum* SK1^T^ (98.4%), *C. soli* GCJ02^T^ (98.4%), *C. lactosi* Sr59^T^ (98.3%), *C. zongtaii* TMN-42^T^ (98.2%), and *C. adonitolivorans* RHLS22-1^T^ (98.1%). The ANI and the DNA–DNA hybridization estimate values between strain 1639^T^ and all type strains of species of the genus *Cryobacterium* were in the range of 84.3–87.8% and 20.5–40.3%, respectively. The combined genotypic and phenotypic data indicate that strain 1639^T^ represents a novel species within the genus *Cryobacterium*, for which the name *Cryobacterium inferilacus* sp. nov. is proposed with the type strain 1639^T^ (=KCTC 59142^T^, =VKM Ac-2907^T^, UQM 41460^T^).

## 1. Introduction

In the course of an ecological study of microorganisms in Antarctica, Inoue isolated an obligately psychrophilic aerobic Gram-positive pleomorphic rod-shaped bacterium from soil of Antarctica, for which the name “*Curtobacterium psychrophilum*” was proposed [1]. The bacterium grew optimally within the range from 9 °C to 12 °C and did not grow at temperatures higher than 18 °C. However, the name was not validated. Moreover, phylogenetic analysis based on 16S ribosomal DNA sequences revealed that this organism was positioned at a separate branch in the family *Microbacteriaceae* in the class *Actinobacteria*. It was allocated to the new genus *Cryobacterium* with *Cryobacterium psychrophilum* gen. nov., sp. nov. as the type species [2]. The genus *Cryobacterium* presently includes 33 species with validly published names, most of which are cold adapted microorganisms isolated from samples taken from low-temperature ecosystems: glaciers. These strains formed pigmented colonies and were found to be Gram-positive, rod-shaped, and catalase-positive bacteria. Among the representatives of this genus, there are psychrophilic, psychrotolerant, and mesophilic species.

Psychrophilic species, along with *C. psychrophilum* (T_range_ 0–18 °C, T_opt_ 9–12 °C), include *C. flavum* (T_range_ 0–19 °C), *C. luteum* (T_range_ 0–20 °C), *C. melibiosiphilum* (T_range_ 0–18 °C, T_opt_ 10–15 °C), and *C. ruanii* (T_range_ 0–18 °C, T_opt_ 10–14 °C), which were isolated from ice samples collected from the ice tongue surface of the No. 1 Glacier in Xinjiang Uygur Autonomous Region [3,4,5]. *C. levicorallinum* (T_range_ 0–18 °C, T_opt_ 8–14 °C), and *C. aureum* (T_range_ 0–18 °C, T_opt_ 8–14 °C) were also isolated from an ice sample collected from the China No.1 glacier in the Xinjiang Uygur Autonomous Region [6,7]. *C. breve* (T_range_ 0–20 °C, T_opt_ 10–14 °C) was isolated from the supraglacial zone of the Tumingmengke Glacier in Gansu Province [5], and *C. roopkundense* (T_range_ 15–18 °C) was isolated from a soil sample taken at the edge of the Rupkund glacial lake, located in the Himalayan Mountains (India) [8]. Bacteria of these species are adapted to low temperatures and do not grow above 20 °C.

Psychrotolerant representatives of the genus *Cryobacterium* are characterized by a wider temperature range of growth from 4 °C to 28 °C. Despite the fact that they are capable of growing at low temperatures, these species have optimum growth temperatures in the range of 18–22 °C. Four such species have been described: *C. psychrotolerans* from frozen soil collected from the China No.1 glacier [9]; *C. arcticum* from a soil sample collected from Store Koldewey, north-east Greenland [10]; *C. zongtaii* from a glacier in China [11]; and *C. soli*, a member of the microbial community of the forest soil of the city of Baishan (China) [12].

Mesophilic *Cryobacterium* species are *C. mesophilum* isolated from the soil of Bigeum Island [13] and *C. tepidiphilum* isolated from the rhizosphere of lettuce [14]. They grow at temperatures in the range of 20–28 °C with an optimum at 28 °C and from 10 °C to 30 °C with an optimal growth temperature at 20–25 °C, respectively.

Liu with co-authors isolated 19 bacterial strains from ice, cryoconite, and meltwater samples in the supraglacial zone of Xinjiang No. 1, Toumingmengke, Hailuogou, and Midui glaciers of China. In 2023, these strains were selected for polyphasic taxonomic analysis and have been described as 19 novel *Cryobacterium* species [15]. Comparative genomic analysis of these strains showed that there are significant differences in the gene content between groups with different Tmax values (≤20 °C and >20 °C). Comparative genomic analysis of genome sequences of 101 strains showed that they could be classified into 44 phylogenetic groups at the species level according to the “genomic gold standard” [16]. It has been shown that some species showed intraspecific diversity. For example, eleven *Cryobacterium* strains were clustered into three branches, and seven *Cryobacterium levitaulinum* strains were divided into three branches [15]. Thus, there is phenotypic and genetic diversity of the genus *Cryobacterium,* a group specific to cryosphere environments.

The purpose of the present work was to describe the morphology, physiology, and chemotaxonomic features of the strain 1639^T^, isolated from a water sample of subglacial low-temperature Lake Untersee (Antarctica), and determine the taxonomic position of a new organism. In this study, we present new information on the phenotype of the ureolytic 1639^T^ strain supplemented by genome sequencing and phylogenomic analysis to support the conclusion of the affiliation of this strain to a new species of the genus *Cryobacterium*.

## 2. Materials and Methods

### 2.1. Home Habitat and Isolation

Strain 1639^T^ was isolated from a water sample, taken from a depth of 75 m with a temperature of 4.0 °C and alkaline pH of the subglacial low-temperature oligotrophic Lake Untersee (East Antarctica: 71°20′ S, 13°45′ E), located in the interior of the Gruber Mountains of central Queen Maud Land. We used basal mineral ATCC medium [17]. To obtain an enrichment culture, it was added to the sample in a ratio of 4:1 (v/v), and the medium was supplemented with yeast extract, tryptone, casaminic acids, and peptone in a concentration of 0.001% each as stimulants of bacterial growth. The sample prepared in this way was incubated for 1 month at 4 °C. To obtain separate bacterial colonies, the enrichment culture was spread on an ATCC agar medium with increased content of yeast extract, tryptone, casaminic acids, and peptone (up to 0.005% each). After 2 weeks of aerobic incubation at 4 °C, separate visible colonies were obtained. A pure culture was obtained from one colony by repeatedly streaking it on agar medium of the same composition. The isolated bacterium was designated as strain 1639^T^.

### 2.2. Phenotypic and Chemotaxonomic Characteristics

The cell morphology and size were examined by phase contrast light microscopy (Olympus CX41, Tokyo, Japan) and transmission electron microscopy (JEM-1400, JEOL, Tokyo, Japan).

The spectrum of substrates used as the carbon and energy sources was examined on the same mineral medium with the addition of 0.005% yeast extract as the growth stimulator and of each test substrate at a concentration of 0.2%. The studied substrates were sugars (glucose, sucrose, lactose, galactose, maltose, fructose, mannose, arabinose, rhamnose, trehalose, xylose, ribose, cellulose, cellobiose, amylose, starch, and xylan); alcohols (ethanol, glycerol, sorbitol, and mannitol); sodium salts of organic acids (acetate, propionate, butyrate, pyruvate, lactate, fumarate, and succinate); amino acids (alanine, serine, glutamine, arginine, proline, cysteine, and tryptophan); C1-compounds (methanol, monomethylamine, dimethylamine, trimethylamine, and urea); gelatin; and cellulose.

Growth was measured by monitoring turbidity at 600 nm twice a day over a period of 2 weeks on a spectrophotometer (UNICO 2100, Dayton, NJ, USA).

The description of the colonies was carried out after growing strain 1639^T^ on agar medium with 0.2% sucrose for 14 days at an optimal growth temperature.

The temperature and pH range for growth were determined by growing strain 1639^T^ in a liquid medium with 0.2% sucrose as a substrate in the temperature range from 0 to 40 °C and pH within the range of 4–10 (pH 4.0–10.0) with citrate/phosphate (pH = 4.0–7.0), Tris/HCl (pH = 7.2–9.0) or sodium carbonate/sodium bicarbonate (pH = 9.0–10.0) buffers).

Growth with NaCl was investigated in the range of 0–10% (*w*/*v*) at intervals of 1%.

Growth on urea as a nitrogen source was investigated by adding NH_4_ to a liquid medium with sucrose as a substrate in a 1:1 ratio. The control was a medium of the same composition with sucrose as a substrate without urea.

Catalase activity was determined by bubble production in 10% (*v*/*v*) hydrogen peroxide solution.

Oxidase activity was analyzed using 1% N, N, N′, N′-tetramethyl-p-phenylenediamine as a redox indicator.

Proteolytic activity was established by the ability to liquefy gelatin. The presence of hydrolytic enzymes was determined by the ability of the strain to hydrolyze starch and cellulose.

Antibiotic sensitivity was studied by using filter discs (Oxoid) on MS plates and measuring the inhibition zones. Eleven antibiotics were investigated: erythromycin (15 µg), streptomycin (10 µg), rifampicin (30 µg), ampicillin (10 µg), penicillin (1.2 µg), novobiocin (30 µg), chloramphenicol (30 µg), kanamycin (30 µg), neomycin (30 µg), gentamicin (10 µg), and lincomycin (15 µg).

For quantitative analysis of the cellular fatty acid composition and membrane lipids, cells were harvested after 5 days of incubation. Cellular fatty acids were determined as described earlier [18].

Bacterial membrane lipids were determined by thin-layer chromatography. The sample for analysis was prepared as follows. The bacterial biomass was triturated with sodium sulfate in isopropanol, after which the lipids were extracted for 30 min at 70 °C and the supernatant was decanted [19]. The residue was then extracted twice with iso-propanol–chloroform (1:1). The combined extract was evaporated on a rotary evaporator, and the residue was dissolved in 9 mL of chloroform–methanol (1:1), to which 12 mL of a 2.5% sodium chloride solution was added to remove water-soluble substances. After separation of the mixture, the chloroform layer was collected and dried with anhydrous sodium sulfate, evaporated on a rotary evaporator, and dried to a constant weight in vacuum. The resulting residue was dissolved in chloroform–methanol (1:1) and stored under −21 °C. Separation of polar lipids was carried out using two-dimensional TLC on glass plates from Merck, (Darmstadt, Germany) in the following solvent system: chloroform–methanol–water (65:25:4) in the first direction; chloroform–acetone–methanol–acetic acid–water (50:20:10:10:5) in the second direction [20]. Next, 100–200 μg of lipids were applied to the plate. Chromatograms were developed by spraying with a solution of 5% sulfuric acid in ethanol, followed by heating at 180 °C until spots developed. To identify the lipids, individual tags and qualitative reactions were used with ninhydrin (for the presence of an amino group), Dragendorff reagent (for choline), and α-naphthol (for carbohydrate groups) [21].

Quinones were extracted from freeze-dried biomass with chloroform–methanol (2:1), purified by TLC with hexane–diethyl ether (85:15) elution, separated with the eluent acetonitrile–isopropyl alcohol (60:40, *v*/*v*) by an Agilent HPLC-ELSD system with an Agilent Eclipse Plus C18 column (4.6 × 250) and analyzed on a ThermoScientific Orbitrap mass spectrometer.

To investigate the cell wall composition, native cell walls were obtained by differential centrifugation after preliminary disruption by sonication of cells (UP100H, Hielscher, Germany, 30 kHz, 3–5 times per 10 min) and purified with 2% sodium dodecyl sulfate (SDS) as described in [22]. To obtain peptidoglycan, teichoic acids and other carbohydrate-containing polymers were removed from the cell walls by extraction of 5% TCA at 100 °C 20 min. Then, 10 mg of trypsin (1 mg/mL) in Tris-HCl buffer at a pH of 7.8 was added to the remains of the cell walls, and enzyme treatment was carried out at 37 °C for 20 h. The obtained preparation was repeatedly washed with distilled water and then treated with 4% SDS in the same buffer at 100 °C for 5 min. After SDS removal by washing, the peptidoglycan preparation was lyophilized. Amino acids of peptidoglycan were determined after sample hydrolysis (freshly prepared concentrated hydrochloric and trifluoroacetic acids in a 2:1 ratio with the addition of 0.1% β-mercaptoethanol, 1 h at 155 °C) on an L-8800 amino acid analyzer (Hitachi, Tokyo, Japan) according to the method described in [23].

### 2.3. 16S rRNA Gene and Genome Sequencing and Annotation

The Power Soil kit (MO BIO Laboratories, Carlsbad, CA, USA) was used to isolate genomic DNA of strain 1639^T^. Primers 27F and 1492R [24] were used to amplify the 16S rRNA gene of strain 1639^T^. Purified PCR products were sequenced on an ABI Prism 3730 DNA analyzer (Applied Biosystems, Foster City, CA, USA) using a Big Dye Terminator reagent kit version 3.1 in accordance with the manufacturer’s recommendations.

The method of isolation of genomic DNA of strain 1639^T^, sequencing conditions, programs for checking the quality of the obtained sequences, and assembling scaffolds from contigs were described by Grouzdev [25]. The average nucleotide identity (ANI) was determined using FastANI v. 1.34 [26]. Digital DNA-DNA hybridization (dDDH) of genomes was performed using the Genome-to-Genome Distance Calculator (GGDC) v. 3.0 [27]. Identification of protein-coding sequences and primary annotation of genes were performed using the NCBI Prokaryotic Genome Automatic Annotation Pipeline (PGAP), version 6.7 [28]. In addition, to identify protein-coding sequences, we used UniProt v. 2024_06 (https://www.uniprot.org/, accessed on 24 July 2024) [29].

The 16S rRNA gene sequences were aligned using MUSCLE version 5.3, 11 November 2024 [30]. Maximum-parsimony and neighbor-joining trees were reconstructed using MEGA11 [31]. A whole-genome-based phylogenetic tree was generated using the CodonTree method within BV-BRC 3.45.4 (https://www.bv-brc.org/, accessed on 1 August 2024) [32], which used PGFams as homology groups. The genomes of type strains of all validated species of the genus *Cryobacterium* were used in the analysis for comparison. A total of 445 PGFams were found among these selected genomes using CodonTree analysis, and the aligned proteins and coding DNA from single-copy genes were used for RAxML analysis [33]. iTOL version 7.0 was used for tree visualization [34].

Pan-genome analysis was performed using the bioinformatic pipeline IPGA (https://nmdc.cn/ipga/, accessed on 24 July 2024) version 1.09 [35]. The integrated dbCAN3 meta server, version 4.1.14, was used with default settings to classify the carbohydrate-active enzyme (CAZyme)-encoding genes in the genome of strain 1639^T^ [36]. The reconstruction of possible metabolic pathways was carried out based on comparison of the genome of the strain 1639^T^ using the BlastKOALA tool of KEGG, version 3.1 (https://www.kegg.jp/blastkoala/, accessed on 1 January 2025) [37]; MetaCyc, version 28.5, (https://metacyc.org/, accessed on 11 December 2024), [38]; RAST v. 2.0 (https://rast.nmpdr.org/rast.cgi, accessed on 12 October 2024) [39]; and BV-BRC. The circular genome map of strain 1639^T^ was constructed using the Proksee web service (https://proksee.ca/, accessed on 1 August 2024) [40]. Gene clusters of urea assimilation and arsenic resistance were drawn using the online service Gene Graphics version 2.02 (https://katlabs.cc/genegraphics/app, accessed on 1 October 2024) [41].

### 2.4. Nucleotide Sequence Accession Numbers

The GenBank/EMBL/DDBJ accession number for the 16S rRNA gene sequences of the strain 1639^T^ is MZ936365. The genome shotgun project of strain 1639^T^ has been deposited at DDBJ/ENA/GenBank under the accession no. JAIEUL000000000. The version described in this paper is the first version, RBWE01000000. The NCBI genomic assembly accession number of strain 1639^T^ is GCF_019599185.1.

## 3. Results and Discussion

### 3.1. Morphological Characteristics

Cells of strains 1639^T^ were Gram-stain-positive, aerobic, non-motile, non-spore-forming, and short pleomorphic rods with dimensions of approximately 0.3–0.5 × 1.2–1.4 µm (Figure 1).

Colonies were small (1.5–2 mm), round, yellow–green, smooth, opaque, and slightly convex with thick consistency.

### 3.2. The Spectrum of Substrates

Strain 1639^T^ was heterotrophic and used a wide range of organic compounds as substrates: methanol, sucrose, cellobiose, amylose, arabinose, ribose, glucose, fructose, maltose, lactose, galactose, xylose, trehalose, pyruvate, acetate, lactate, propionate, and urea. The addition of urea to the medium with sucrose as a substrate increased the growth of the culture by 1.5 times under experimental conditions.

The cells of the new bacterium did not grow on monomethylamine, dimethylamine, trimethylamine, rhamnose, mannose, xylan, cellulose, ethanol, glycerol, sorbitol, and mannitol; sodium salts of butyrate, fumarate, and succinate; as well as the amino acids alanine, serine, arginine, glutamine, proline, cysteine, and tryptophan. It could not hydrolyze gelatin, cellulose, and starch. Since sucrose was the preferred substrate, studies of other characteristics of strain 1639^T^ were carried out using this substrate.

### 3.3. Phenotypic Characteristics

Strain 1639^T^ could grow between 0 and 25 °C (with Topt at 10 °C) at a pH between 4.5 and 8.7 with the optimum at 6.8–7.8.

Strain 1639^T^ was capable of growth in a wide range of NaCl concentrations up to 2%.

The strain showed a positive result for catalase but a negative one for oxidase.

The cells did not have proteolytic activity, as they did not dilute gelatin.

The activity of hydrolytic enzymes was not established, since strain 1639^T^ did not hydrolyze cellulose.

Strain 1639^T^ was sensitive to erythromycin, neomycin, streptomycin, rifampicin, ampicillin, novobiocin and resistant to penicillin, chloramphenicol, kanamycin, gentamicin, and lincomycin.

The major fatty acids of strain 1639^T^ were i-C_16:0_ (49.69%), ai-C_15:0_ (17.59%), and C_16:1 branched_ (12.03%). The cells also contained C_15:1 branched_ (7.81%), ai-C_17:0_ (6.14%), C_15:1 branched_ (3.67%), C_16:0_ (2.29%), C1_8:0_ (0.62%), and C_20:0_ (0.16%).

Identified polar lipids were phosphatidylglycerol, unidentified phospholipids, and glycolipids.

Only one menaquinone, MK-10, was detected.

Amino acid analysis of the peptidoglycan preparation showed the presence of 2,4-diaminobutyric acid and ornithine, glycine, alanine, and glutamic acid in the ratio 1.0:0.9:0.6:1.0.

Strain 1639^T^ had some phenotypic differences from closely related species of the genus *Cryobacterium* (Table 1).

The organism was psychrophilic with an optimal growth temperature of 10 °C. It did not exhibit mobility. Unlike closely related species, strain 1639^T^ had a number of differences in the use of substrates for growth, including the use of methanol. The main difference from other closely related representatives of the genus *Cryobacterium* was the ability to use urea as a source of nitrogen. Strain 1639^T^ had ornithine in the cell wall. The new organism contained monounsaturated fatty acid (C_16:1_ branched) as the dominant one.

Polar lipids of the strain 1639^T^ were phosphatidylglycerol and unidentified glycolipids. Diphosphatidylglycerol was not found in cells.

### 3.4. Genome Analysis

The genome of the strain 1639^T^ was analyzed, and the indices of the genomic relationship were determined to clarify its taxonomic position taking into account the present time approaches in order to reveal the potential metabolic functions, which may by not detected by phenotypic studies. Genome analysis made it possible to correlate the genomic information and phenotypic features of the strain 1639^T^ and to give a formal description of the new species.

### 3.5. Phylogenetic Analysis of the 16S rRNA Gene

The sequence of the amplified 16S rRNA gene region of strain 1639^T^, 1445 nucleotides long, was identical to the partial sequence of the 16S rRNA gene annotated in the genome of this strain. Phylogenetic analysis of 16S rRNA gene sequences in BLAST showed that the strain 1639^T^ showed 98.4%, 98.4%, 98.3%, 98.2%, and 98.1% similarity with the respective genes of the nearest members of the genus *Cryobacterium*: *C. arcticum* SK1^T^, *C. soli* GCJ02^T^, *C. lactosi* Sr59^T^, *C. zongtaii* TMN-42^T^, and *C. adonitolivorans* RHLS22-1^T^. On the phylogenetic tree constructed using the NJ and ML methods, strain 1639^T^ formed a separate branch within a well-supported clade of phylogenetically closely related species, including *C. arcticum* SK1^T^, *C. soli* GCJ02^T^, *C. lactosi* Sr59^T^, *C. zongtaii* TMN-42^T^, and *C. adonitolivorans* RHLS22-1^T^ (Figure 2), which, according to an earlier phylogenomic analysis [15], was designated as group 2 of moderately psychrophilic bacteria of the genus *Cryobacterium*.

This indicated that the studied strain 1639^T^ belonged to the genus *Cryobacterium* but did not belong to any of the known species of this genus. Importantly, when the trees were constructed using the neighbor-joining and maximum-likelihood algorithms, the genus *Cryobacterium* appears to be polyphyletic, suggesting that further taxonomic revision may be required to fully resolve the evolutionary relationships within this group.

### 3.6. Genome Statistics, Pan-Genome, and Phylogenetic Analysis

The genome of strain 1639^T^ was composed of 75 scaffolds with a total genomic length of 4,017,491 bp, an N50 value of 150.3 kb, a G + C content of 68.03%, and coverage of 234.2×. The genome completeness and contamination were assessed as 98.05% and 2.89%, respectively. The genome comprised 3793 annotated genes, including 3686 protein-coding sequences, 50 pseudogenes, and 57 RNA genes. The ANI and dDDH values of the 1639^T^ genome with the genomes of the type strains of the genus *Cryobacterium* were in the range of 84.3–87.8% and 20.5–40.3%, respectively (Appendix A).

These values are below the thresholds of 95–96% for ANI and 70% for dDDH accepted for assigning bacteria to a single species [42,43], which indicates that strain 1639^T^ belongs to a new species.

Genomes of all 34 type strains of *Cryobacterium* species and of strain 1639^T^ were used for pan-genome analysis. To identify the phylogenetic position of the strain, phylogenomic trees were constructed based on concatenated 445 single-copy proteins using the Bacterial Genome Tree Service of the BV-BRC portal (Figure 3) and also based on genome sequences, single-copy core gene sequences, and ANI values using the IPGA portal (Appendix A).

On all phylogenomic trees, strain 1639^T^ formed a separate branch in the same cluster as on the “ribosomal” tree with the type strains of *C. arcticum* SK1^T^, *C. soli* GCJ02^T^, *C. lactosi* Sr59^T^, *C. zongtaii* TMN-42^T^, and *C. adonitolivorans* RHLS22-1^T^ species, which also confirms the assumption that this strain could be assigned to a new species.

When the genome of strain 1639^T^ was added to the pan-genomic analysis of 34 genomes of typical strains of *Cryobacterium* species, the number of pan-gene clusters (GCs) increased to 30,915 and core GCs decreased to 729, but the curve gradually flattened out (Appendix A).

At the same time, among all orthologous GCs, core genes accounted for only 2.42%, and the number of unique genes belonging to different genomes rose to 55.76 (Appendix A). The amount of unique genes in each genome was in the range of 248–1161 (Appendix A). Among them, there were more unique genes for strain 1639^T^ (632) than for the strains *C. zongtaii* TMN-42^T^ (440) and *C. adonitolivorans* RHLS22-1^T^ (335), belonging to the same close phylogenetic cluster, as well as some other type strains of the genus *Cryobacterium*. These results all provided gene-level evidence indicating that strain 1639T is highly divergent from other *Cryobacterium* species and may receive the status of a separate species. The primary COG annotation showed that 632 unique genes in the pan-genome result for strains 1639^T^ included 70 genes for metabolism, 60 for cellular processes and signaling, 33 for information storage and processing, and 461 for not annotated and poorly characterized. But only 30 of these GCs unique to other *Cryobacterium* species had a function predicted according to Prokka, KEGG, and RAST annotations.

This unique gene set underscores the specialized adaptations of strain 1639^T^, particularly in nutrient uptake, DNA damage, and defense mechanisms. As for the experimentally discovered ability of strain 1639^T^ to grow with urea, which is rare for this genus, two gene complexes (*ure*ABCFGD and *uca-atz*F) involved in urea assimilation together with a complex of unique genes of urea transport have been annotated in its genome (*urt*ABCDE). The genome also harbors two unique gene clusters for arsenic resistance (*ars*RC1ADCCCBR and *ars*ADCCB). The unique *umu*DC operon encodes a protein that, together with the product of the *rec*A gene, can carry out DNA-damage-induced mutagenesis, i.e., SOS mutagenesis. This machinery allows the replication to continue through DNA lesion, and therefore avoid lethal interruption of DNA replication after DNA damage [44]. The Crp-Fnr regulator (unique gene *fnr*) stands out in responding to a broad spectrum of intracellular and exogenous signals such as anoxia, the redox state, oxidative stress, or temperature [45]. The unique *lnr*LMN operon encodes a multifunctional ABC transporter involved in dual antibiotic resistance and biofilm morphology, contributing to increased fitness [46]. Two *tri*A unique genes encoding melamine deaminase displaced two of the three amino groups from melamine, producing ammeline and ammelide as sequential products [47].

The circular map of genome contigs of the strain 1639^T^ (Figure 4) obtained using the Proksee server, shows localization of the unique single genes and gene operons presumably identified by pan-genomic analysis carried out in this work. 

For comparison with the genome of strain 1639T, the map shows data from BLAST analysis with the genomes of five phylogenetically similar strains (C. zongtaii TMN-42T—PPXD00000000; C. arcticum SK1T—GHLY00000000; C. adonitolivorans RHLS22-1T—SOFL00000000; C. lactosi Sr59T—SOHM00000000; C. soli GCJ02T—CP030033). The results of this comparison show that the genome of strain 1639T contains regions that are not represented in the genomes of other similar strains. In most cases, these sites include unannotated and poorly characterized unique genes. At the same time, some of the annotated unique genes belong to such sites, and some of them, according to the results of the built-in Alien Hunter module, were presumably obtained via horizontal gene transfer (Appendix A).

This genomic and pan-genomic characterization of the strain 1639^T^ enriches our understanding of its unique physiological capabilities and potential environmental adaptations, contributing to a broader understanding of the physiological diversity and evolutionary strategy of the *Cryobacterium* genus.

### 3.7. Genome Functional Characterization

Functional annotation revealed that out of the total 3686 protein-coding genes from the 1639^T^ genome, 2947 classified into Clusters of Orthologous Genes (COG) functional categories (Figure 5A).

The top three functional COG terms were carbohydrate transport and metabolism, G (12%); amino acid transport and metabolism, E (10%); and transcription, K (10%). A total of 2356 genes from the genome of strain 1639^T^ were annotated in the Gene Ontology (GO) database to 4801 GO terms under the three broad categories of biological process (BP), cellular component (CC), and molecular function (MF) (Figure 5C). The top three annotated BPs were the metabolic processes (52%), cellular processes (50%) and biological regulation (14%). The top three CCs included cell (33%), cell part (33%), and membrane (21%). The top annotated MFs included catalytic activity (63%), binding (44%), and transporter activity (10%). According to the Kyoto Encyclopedia of Genes and Genomes (KEGG) 3618 genes of the 1639^T^ genome were successfully annotated with 1625 KEGG Orthologous (KO) terms (Figure 5B). In the category of metabolic processes, the genes of carbohydrate metabolism (274) and amino acid metabolism (227) pathways were the most numerous. In the genetic information processing category, the translation pathway had the most annotations of 77 genes. In the environmental information processing category, 120 and 53 genes were enriched in the membrane transport and signal transduction pathways, respectively.

The analysis of the enzyme composition of the metabolic pathways was performed based on the results of functional prediction of proteins using the BlastKOALA, BV-BRC, RAST, and UniProt services. The genome of strain 1639^T^ presumably contains all the genes responsible for the complete modules of pathways including the central pathways of carbohydrate metabolism: glycolysis (Embden-Meyerhoff pathway) and glucogenesis, pyruvate oxidation, citrate cycle (TCA cycle, Krebs cycle), pentose phosphate pathway (pentose phosphate cycle), and 5-phospho-alpha-D-ribose 1-diphosphatefructose (PRPP) biosynthesis. For other pathways of carbohydrate metabolism and nucleotide sugar biosynthesis, the genes of galactose degradation (Leloir pathway), glyoxylate cycle, methylcitrate cycle, UDP-glucose, UDP-galactose, UDP-N-acetyl-D-glucosamine, and dTDP-L-rhamnose biosynthesis were annotated. In addition, a complete set of genes for enzyme complexes of ATP synthesis has also been annotated: succinate dehydrogenase, cytochrome bc1 complex respiratory unit, F-type ATPase (EC: 7.1.2.2), and cytochrome c oxidase. Other pathways for which complete gene sets were annotated include dissimilatory nitrate reduction, formaldehyde assimilation, ribulose monophosphate pathway, phosphate acetyltransferase-acetate kinase pathway, base amino acids biosynthesis, ornithine biosynthesis and ornithine-ammonium cycle, beta-oxidation of fatty acids, de novo purine biosynthesis, and biosynthesis of vitamins and cofactors as follows: thiamine (thiamine salvage pathway), riboflavin and FAD, pyridoxal-P, NAD, pantothenate and coenzyme A, lipoic acids, and siroheme and heme in porphyrin metabolism. No genes of complete pathways for xenobiotic degradation were found.

### 3.8. Carbohydrate Metabolism and Oxidation of Organic Compounds

Carbohydrate metabolism is a crucial part of the life cycle of bacteria, ensuring their survival in various habitats. Enzymes that catalyze the synthesis and breakdown of glycosidic bonds account for 1–3% of the proteins encoded by the genomes of most organisms [48]. Genome-wide screening for carbohydrate-active enzymes (CAZymes) by the dbCAN3 server in the type strains of 35 *Cryobacterium* species, including the new strain 1639^T^, retrieved various functional classes suggesting the ability of the analyzed members of the genus *Cryobacterium* to metabolize different carbohydrates: glycoside hydrolases (GHs), glycosyltransferases (GTs), polysaccharide lyases (PLs), carbohydrate esterases (CEs), carbohydrate-binding modules (CBMs), and auxiliary activities (AAs). On average, the number of GH and GT families of CAZymes in the phylogenetic cluster of strain 1639^T^ and related strains was higher than in other species of the genus *Cryobacterium* (Appendix A).

The total annotated CAZyme gene number in the 1639^T^ genome was 124, including 54 GHs, 50 GTs, 8 CEs, 8 AAs, and 3 CBMs. No gene was assigned to PLs. The representation of various GH families in strain 1639^T^ differed slightly from that of related strains, mainly combining glucosidase, i.e., enzymes hydrolyzing O- and S-glycosyl compounds (3.2.1.-). The GH13 family was the most extensive, which includes the most typical and studied enzyme α-amylase (EC: 3.2.1.1), specifically catalyzing the hydrolysis of α-1,4-glycosidic linkages of starch to produce small molecular products including glucose, maltose, and maltotriose [49]. However, in the genome of strain 1639^T^, the *alm*A of α-amylase genes is not unambiguously annotated, which corresponds to its experimentally detected inability to hydrolyze starch. At the same time, the *tre*X genes of isoamylase (EC 3.2.1.68, GH13 family) were annotated in its genome, as well as the *chi*C gene of chitinase (EC 3.2.1.14, GH18 family), which indicate the possible ability to degrade glycogen and chitin, respectively.

Compared with most related reference strains, strain 1639^T^ lost only the GH127 family, including beta-L-arabinofuranosidase (3.2.1.185), the enzyme that was identified in the bacterium *Bifidobacterium longum* JCM1217, removing β-L-arabinofuranose residue from the non-reducing end of multiple substrates [50]. At the same time, the subfamily GH13-13 with the rare enzyme pullulanase (EC: 3.2.1.41) was detected only in strain 1639^T^. This enzyme is encoded by the supposedly annotated *pul*A gene in the genome of strain 1639^T^; it hydrolyzes either α-1,6 and α-1,4 or both glycosidic bonds in pullulan as well as in other carbohydrates (amylopectin and glycogen) to produce glucose, maltose, and maltotriose syrups, which have important uses in the food industry and other related sectors [51].

The highest number of glycosyltransferase (GT) families (50) was predicted for strain 1639^T^ among all 34 typical strains of the genus *Cryobacterium*. However, the composition of GT families of this and related strains differed slightly, although the GT9 family, comprising enzymes with two known activities, lipopolysaccharide N-acetylglucosaminyltransferase (EC: 2.4.1.56) and heptosyltransferase (2.4.-.-), was specific only for strain 1639^T^.

The ability of strain 1639^T^ to use a wide range of mono-, di-, oligo-, and polysaccharides as substrates correlates with the results of its genome analysis. According to the results of the BlastKOALA and RAST services, the genes for glucose (*pg*i, *pgk, fba*A, *eno*, *pfk*A, *gap*, *pyk*, and *gpm*A); sucrose (*mal*Z and *sac*A); cellobiose (*bg*lB); D-fruktose (*fru*A, *fru*K and *scr*K); amylose (*glg*B); L-arabinose (*ara*A, *ara*B, *ara*D and *rpe*); maltose (*mal*Z); lactose (*lac*Z); D-galactose (*gal*M, *gal*K, and *gal*T); D-ribose (*rbs*K); D-xylose (*xyl*A, *xyl*B, and *rpe*); and trehalose (*tre*P and *tre*A) consumption were annotated in the genome of strain 1639^T^.

The genes of enzymes for the oxidation of the organic substrates pyruvate (*ace*E, *pdh*D and *pdh*C), acetate (*ask*A and *pta*), L-lactate (*lld*D and *ldh*), and propanoate (asc, *prp*C, *prp*D, *acn*A, and *prp*B) are also presumably annotated in the genome of strain 1639^T^.

Although the genes of specific methanol degradation enzymes in the genome of strain 1639^T^ have not been annotated, its ability to use methanol discovered in the experiment may be related to the functioning of the first stage of the ADH pathway, based on the annotation in its genome of alcohol dehydrogenase enzyme ADH genes (EC: 1.1.1.1 and 1.1.1.2) capable of oxidizing methanol to formaldehyde, which is then catabolized with the participation of the genes of formaldehyde assimilation (*hxl*A, *hxl*B, *pfk*, and *fba*A).

### 3.9. Urea Metabolism

The experimentally established ability of strain 1639^T^ to use extracellular urea as a nitrogen source is a unique property for members of the genus *Cryobacterium*. In other *Cryobacterium* species, this property is either absent or has not been determined.

The most widely known is the urease pathway, widespread among bacteria, fungi, and plants. Urease (EC: 3.5.1.5), a metalloenzyme with two nickel ions in its active center [52], catalyzes the hydrolysis of urea to ammonia and carbonic acid. Although urease gene clusters differ for different bacteria, they usually include structural and accessory genes. In the genome of strain 1639^T^, the urease gene cluster (K2F54_10550-10580) consists of structural *ure*ABC genes (K2F54_10570-10580) encoding three subunits of urease and accessory *ure*FGD genes (K2F54_10555-10565) encoding the proteins that carry out the metal assembly process between nickel ions and urease structural proteins. In addition, the entry of extracellular nickel ions, which are necessary for the functioning of urease, is facilitated by the transmembrane transporter of nickel ions, encoded by the *nix*A gene (K2F54_10550). This cluster is flanked by the *ars*R gene of metalloregulator ArsR family transcription factor (K2F54_10545). Extracellular urea uptake occurs via ABC-type (ATP-binding cassette) transporters that use energy from ATP to transport urea across the cell membrane and are encoded by the *urt*ABCDE cluster (K2F54_10585-10605). This cluster is flanked by the *tet*R genes of the TetR family protein (K2F54_10585-10610) that can control gene expression depending on the availability of nitrogen sources (Figure 6).

In the genomes of the type strains of most *Cryobacterium* species, urease genes are not annotated except for the genome of *C.psychrophylum* DSM 4854^T^, which contains the structural and auxiliary genes of the urease operon. However, the cluster of ABC-type (ATP-binding cassette) transporter genes in its genome is not annotated, so the possibility of using extracellular urea by this strain is doubtful.

The alternative pathway for urea degradation occurs via urea amidolyase (EC 3.5.1.45) and involves the activities of biotin-dependent urea carboxylase and allophanate hydrolase. Urea amidolase catalyzes the ATP- and biotin-dependent hydrolysis of urea to ammonia and carbonic acid with allophanate as an intermediate. For a long time, this pathway, encoded by a bifunctional gene, was considered to be realized only in eukaryotes that had lost urease. However, data on the detection of segregated urea carboxylase and allophanate hydrolase activities in the alpha-proteobacterium *Oleomonas sagaranensis* were subsequently obtained, in addition to urease activity [53]. In the genome of strain 1639^T^, a gene cluster (K2F54_17525-17540) was annotated under this study, including the structural genes *uca* and *atz*F, presumably encoding urea carboxylase (EC 6.3.4.6) and allophanate hydrolase (EC 3.5.1.54), respectively, as well as two additional genes, presumably urea carboxylase-related aminomethyltransferase (Figure 6).

The question of the expression of the urea carboxylase genes remains unclear, since the biotin synthase gene (EC 2.8.1.6), which is the key for biotin synthesis, is not annotated in its genome, as well as other genes of this biosynthesis. However, a cluster of genes necessary for transmembrane biotin transport has been annotated in the genome of strain 1639^T^: *bio*YM (K2F54_06215-06220), encoding the substrate-specific component BioY and the ATPase component of energizing module BioM of biotin ECF transporter [54], together with the *bir*A gene (K2F54_07310) of biotin–protein ligase (EC 6.3.4.15), for activating biotin to form biotinyl 5′ adenylate and transferring biotin to biotin-accepting proteins [55]. The presence of these genes presumably allows strain 1639^T^ to use extracellular biotin for the functioning of urea carboxylase. When compared with the genomes of the type strains of *Cryobacterium* species, homologous clusters of urea amidolyase genes were found only in the genomes of *C.soli* and *C.lactosi* from the closely related phylogenetic group, in which the urease cluster genes were not annotated.

Although the study of urea assimilation enzymes by strain 1639^T^ requires special research, it can be noted that the presence of two gene clusters in its genome, presumably determining both known urea assimilation pathways, is unique for the genus *Cryobacterium* and supports classification of the studied strain as member of a new species.

The presence of two genetic systems of urea assimilation in the genome of a single microorganism was also found in the genomes of bacteria of various phyla, including alpha-, beta-, gamma-, and epsilon- proteobacteria, as well as actinobacteria [53]. It was assumed that both pathways can function selectively depending on the concentration of necessary cofactors in the habitat, or one of the pathways can function for other acyl amide decomposition [56]. For strain 1639^T^, it can be assumed that the conditions prevailing in its unique habitat, i.e., in the oxycline of a subglacial lake, in which other nitrogen sources, nitrates + nitrites (<0.04 µm/L), and ammonium (4.11 µm/L) are limited at this depth, affect the expression of genes for urea assimilation pathways [57].

### 3.10. Arsenic Resistance

Arsenic is a toxic metalloid that occurs naturally in aquatic and terrestrial environments in four different oxidation states, As^+5^, As^+3^, As^0^, and As^−3^, of which the most toxic and the most common are arsenate As(V) and arsenite As(III). Many prokaryotes can reduce As(V) to As(III) as a mechanism of resistance to compounds of this toxic metalloid. These arsenate-resistant bacteria do not receive energy from the process but use it as a means of combating high arsenic content in cells. Reduction of arsenate trapped in the cytoplasm of the microorganism to arsenite is mediated by a small polypeptide (ArsC) it is then expelled out of the cell by an As(III)-specific transporter (ArsB) [58]. In general, the *ars* operon, which combines arsenic resistance genes, is detected in the genomes of most bacteria and archaea, localized both on chromosomes and on plasmids and transposons. However, the composition of *ars* genes and their combination in the operon vary significantly in different prokaryotic genomes [59]. The genes of arsenate resistance are represented in the genome of strain 1639^T^ by two *ars* operons. The first large operon, *ars*RC1ADCCCBR (K2F54_15565-15515), comprises the tandem of structural *ars*CCC genes (K2F54_15525-15535) encoding thioredoxin-coupled arsenate reductase (EC: 1.20.4.4), which is common for Gram-positive bacteria. The accessory *ars*A (K2F54_15545) and *ars*B (K2F54_15520) genes, encoding arsenite/antimonite pump-driving ATPase ArsA (EC 3.6.3.16) and, together with the *ars*D gene (K2F54_15540), encoding arsenic metallochaperone ArsD, transfer trivalent metalloids to the ArsAB pump. Another structural gene is *ars*C1 (K2F54_15550), encoding arsenate–mycothiol transferase (EC 2.8.4.2), which, together with mycoredoxin (EC: 1.20.4.3), also reduces arsenate to arsenite [60]. The *mrx*1 gene (K2F54_08575) encoding mycoredoxin is annotated in the 1639^T^ genome outside the *ars* operons. Two more genes, annotated as *ars*C1, are localized in other regions of the genome (K2F54_12335 and K2F54_12335). This particular arsenate reduction pathway was first described for *Corynebacterium glutamicum* and is specific for actinobacteria [60]. The large *ars* operon is flanked by *ars*R genes (K2F54_15515 and K2F54_15565) encoding a metalloregulatory protein ArsR/SmtB family transcription factor. Two additional genes were linked to the *ars* operon: *trx*B (K2F54_15510), encoding thioredoxin reductase (EC 1.8.1.9), which is responsible for reducing the oxidized form of thioredoxin [61], and *ars*M (K2F54_15505), encoding ArsM, an As(III)S-adenosylmethionine methyl transferase enzyme. Arsenic methylation is generally thought of as a detoxification process and is thought to affect the toxicity and availability of arsenic in the environment [62,63]. The second small *ars* operon (K2F54_08885-08905) of the studied genome combines only *ars*ADCCB genes (Figure 7).

Although *ars*C arsenate reductase genes are present in the genomes of type strains of all *Cryobacterium* species, the composition and combination of *ars* operon genes from the genome of strain 1639^T^ are unique. In particular, *ars*C1 genes of arsenate–mycothiol transferase were annotated only in the genomes of type strains of *C*. *arcticum*, *C. zongtaii,* and *C. frigoriphilum*. It is possible that such differences in the organization of the genetic determinants of arsenate resistance in the genomes of even closely related bacterial species are the result of acts of horizontal transfer and translocation of genes associated with arsenate detoxification. Due to the discovery of numerous cases of localization of these genes on plasmids and transposons, the transfer of *ars* genes is considered very likely [59]. It can be noted that the possibility of such a transfer to the genome of strain 1639^T^ is not excluded, since a large *ars* cluster is localized in the area presumably involved in gene transfer (Appendix A).

The detection of genes for arsenate reduction in the genome of strain 1639T suggests its participation in arsenate detoxification under natural conditions, as well as its high biotechnological potential for use in arsenic bioremediation processes, especially under low-temperature conditions.

### 3.11. Cold Stress Protection Genes

Cold is a physical stress that significantly affects all the physical and chemical parameters of a living cell and requires appropriate protective mechanisms against the negative effects of low temperatures on cell structures and metabolic processes [64,65]. For strain 1639^T^, the adaptability to cold conditions was also analyzed at the genome level. The results of the search for the genes responsible for protection against cold shock are shown in Table 2.

Low temperatures affect the DNA structure, causing negative supercoiling. Nucleoid-associated proteins such as gyraseA (*gyr*A) and DNA-binding proteins such as HU-beta (*hup*B) can be used to relax the DNA supercoiling in the genome of strain 1639^T^ [64]. Moreover, several copies belonged to the genes of the CspA family (*csp*A), cold shock proteins that act as chaperones of nucleic acids to prevent the formation of secondary mRNA structures at low temperatures and contribute to the initiation of translation [66]. After translation, DEAD RNA helicases (*csh*A) have been implicated in the stabilization and degradation of mRNAs [67]. Peptidyl prolyl cis-trans isomearase (*cyp*B), which is responsible for protein folding, also seems important for cold adaptation [68] Among the genes encoding ribosome-associated proteins, which are translated during cold stress in strain 1639^T^, are the genes encoding the synthesis of transcription initiation factor I (*inf*A) and ribosome binding factor A (*rbf*A). Other genes implicated in adaptation to cold stress, such as *pun*P encoding purine nucleoside phosphorylase PNP, implicated in RNA degradation, and *ace*E, *pdh*A, *pdh*B, and *pdh*C, encoding pyruvate metabolism enzymes (pyruvate dehydrogenases) [64], were also revealed in the genome of the 1639^T^ strain. Although the genome of strain 1639^T^ does not contain genes for the complete biosynthesis pathway of betaine, a well-known cryo- and osmoprotectant, it does contain genes (*pro*X, *pro*V and *pro*W) of the osmo- and cryostress-protective uptake system of glycine betaine and choline from the environment [69]. Strain 1639^T^ forms green–yellow pigmented colonies; it is possibly due to its ability to synthesize carotenoids. It has been demonstrated that an increase in carotenoid content contributes to membrane stabilization in response to low temperatures [70]. The genome of strain 1639^T^ was found to harbor genes involved in pigment biosynthesis pathways: phytoene synthase (*crt*B) and phytoene desaturase (*crt*I). Thus, in the genome of the Antarctic strain 1639^T^, there is a wide range of genes in different functional categories that contribute to the adaptation of bacteria to low-temperature conditions.

Based on the existing phenotypic and genotypic differences, strain 1639^T^ represents a new species of the genus *Cryobacterium*, for which the name *Cryobacterium inferilacus* sp. nov. is proposed.

## 4. Conclusions

*Cryobacterium inferilacus* strain 1639^T^, which was isolated from a water sample of the subglacial Lake Untersee from a depth of 75 m (oxycline horizon), is the first organism of the genus *Cryobacterium* that uses ammonium nitrogen as a nitrogen source. According to Andersen, the amount of ammonium nitrogen (NH4-N) at the depth of 85–95 m is 12 200–28,000 mg m^−3^. Such an influx of organic matter can be provided by microbial mats, which cover all areas of sediment observed to depths of at least 99 m [71]. Anaerobic decomposition of organic matter may be indicated by a decrease in the amount of nitrate and sulfate and an increase in sulfide from a depth of 85 m, while at 70–75 m, their concentration numbers are H2S—<0.005 (g m^−3^); NO_3_-N—90–17 (mg m^−3^); SO_4_—162.6–159.2 (g m^−3^). Then, at 85 m, NO_3_-N decreases to 1 (mg m^−3^) and SO4 decreases to 22.8 (g m^−3^), while H2S increases to 21.45 (g m^−3^). The amount of ammonium nitrogen in the lake water decreases in the direction from the bottom to the surface along the lake profile. A decrease in ammonium nitrogen begins with a depth of 70–75 m to values of 210–65 (mg m^−3^), and up to the surface of the water, it is at the level of 3–4 (mg m^−3^). Thus, at a depth of 70–75 m, ammonium nitrogen is consumed. At the same time, strain 1639^T^ is able, under conditions of limited (or a lack of) ammonium nitrogen, to use urea produced in the external environment by other microorganisms as an additional nitrogen source. *Cryobacterium inferilacus* strain 1639^T^ is one of the representatives of the microbial community that plays an important role in the consumption of ammonium nitrogen coming from depths <85 m in Lake Untersee, where microbial mats are found.

## 5. Description of Cryobacterium *Inferilacus* sp. nov.

*Cryobacterium inferilacus* (in.fe.ri.la’cus. L. masc. adj. inferus, below, lower; L. masc. n. lacus, lake; N.L. gen. n. inferilacus, of the lower lake, referring to Lake Untersee, Antarctica)

The cells are Gram-stain-positive, aerobic, non-motile, non-spore-forming, and short pleomorphic rods approximately 0.3–0.5 × 1.2–1.4 µm. Colonies are small (1.5–2 mm in diameter) after 14 of days incubation on LMM plates at 10 °C, round, yellow–green, smooth, opaque, and slightly convex with thick consistency. The organism is psychrophilic. Growth occurs at 0–25 °C with the optimum at 10 °C, at pH 4.5–8.7 with the optimum at 6.8–7.8, and in the presence of 0–2.0% (*w*/*v*) NaCl. It is positive for catalase but negative for oxidase. The organism is heterotrophic and uses methanol, sucrose, cellobiose, amylose, arabinose, glucose, fructose, maltose, lactose, galactose, ribose, xylose, trehalose, pyruvate, acetate, lactate, propionate, and urea. The cells did not grow on monomethylamine, dimethylamine, trimethylamine, rhamnose, xylan, cellulose, ethanol, glycerol, sorbitol, mannitol; sodium salts of butyrate, fumarate, and succinate, as well as the following amino acids: alanine, serine, arginine, proline, cysteine, and tryptophan. It cannot hydrolyze, gelatin, cellulose, or starch. The major fatty acids are i-C_16:0_ (49.69%), ai-C_15:0_ (17.59%), and branched C_16:1_ (12.03%). Identified polar lipids were phosphatidylglycerols and glycolipids. The respiratory quinone is MK-10. Amino acid analysis of peptidoglycan preparation showed the presence of ornithine, glycine, alanine, and glutamic acid in the ratio 1.0:0.9:0.6:1.0. The organism is sensitive to erythromycin, neomycin, streptomycin, rifampicin, ampicillin, and novobiocin and resistant to penicillin, chloramphenicol, kanamycin, gentomycin, and lincomycin. The type strain 1639^T^ (=KCTC 59142^T^, =VKM Ac-2907^T^, UQM 41460^T^) was isolated from a water sample, taken from a depth of 75 m in the subglacial low-temperature oligotrophic Lake Untersee, in East Antarctica (71◦20′ S, 13◦45′ E), located in the interior of the Gruber Mountains of central Queen Maud Land. The DNA G+C content of the type strain is 68.03 mol%.

The GenBank/EMBL/DDBJ accession number for the 16S rRNA gene sequences of strain 1639^T^ is MZ936365. The genome shotgun project of strain 1639^T^ has been deposited at DDBJ/ENA/GenBank under the accession no. JAIEUL000000000. The version described in this paper is the first version, RBWE01000000. The NCBI genomic assembly accession number of strain 1639^T^ is GCF_019599185.1.

## Figures and Tables

**Figure 1 microorganisms-13-00990-f001:**
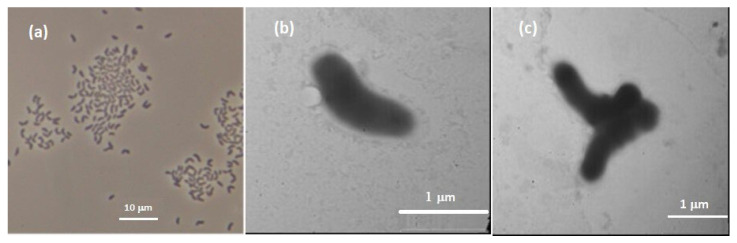
Micrograph in phase contrast (**a**) and transmission electron micrograph of cells of strain 1639^T^ (**b**,**c**). Cells were grown in liquid ATCC at 10 °C for 7 days.

**Figure 2 microorganisms-13-00990-f002:**
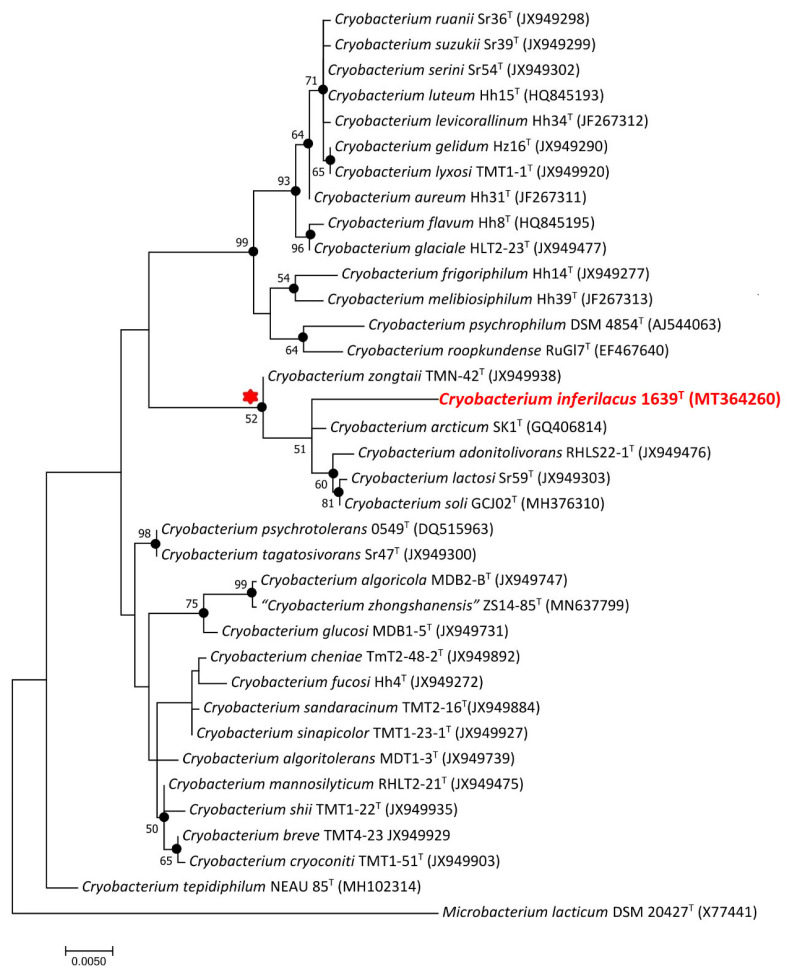
Maximum-likelihood phylogenetic tree based on the 16S rRNA gene sequences (1390 nucleotide sites) showing the position of strains 1639^T^ within the genus *Cryobacterium*. Black circles indicate that the relevant nodes were also recovered based on the maximum-parsimony and neighbor-joining algorithms. Bootstrap values (>50%) are listed as percentages at the branching points. Bar: 0.005 substitutions per nucleotide position. The tree was rooted using *Microbacterium lacticum* DSM 20427^T^ as outgroup. GenBank accession numbers for 16S rRNA genes are indicated in brackets. The name of the strain described in this study is marked by boldface and red. The cluster of species closest to this strain is marked by a red asterisk.

**Figure 3 microorganisms-13-00990-f003:**
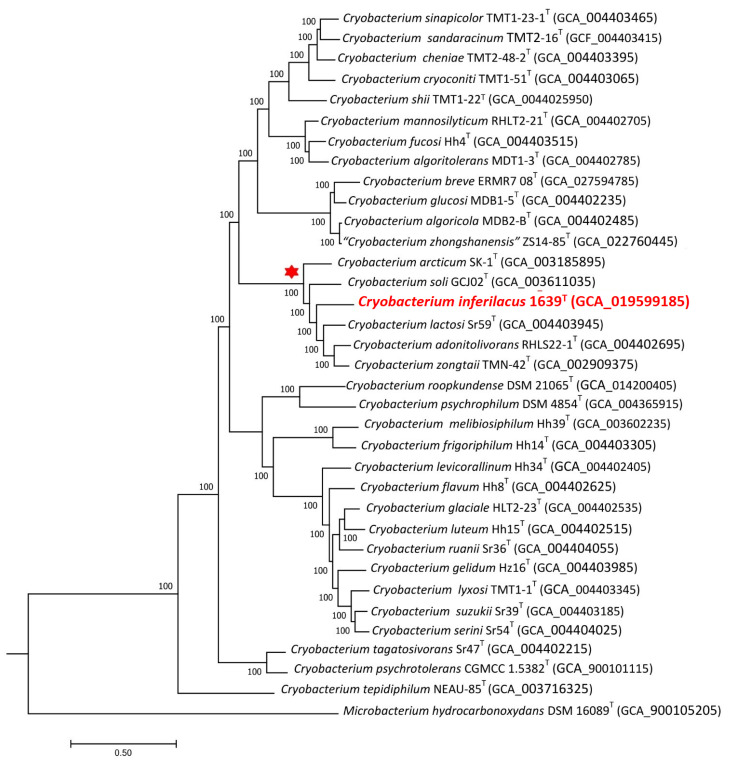
The maximum-likelihood phylogenetic tree derived from 447 single-copy proteins shows the position of strain 1639^T^ within the genus *Cryobacterium*. Bar: 0.05 amino acid substitutions per site. Bootstrap values are listed as percentages at the branching points. The tree was rooted using *Microbacterium hydrocarbonoxydans* DSM 16089^T^ as the outgroup. Accession numbers for the genomic assemblies are indicated in brackets. The name of the studied strain is marked by red boldface. The cluster of species closest to this strain is marked by a red asterisk.

**Figure 4 microorganisms-13-00990-f004:**
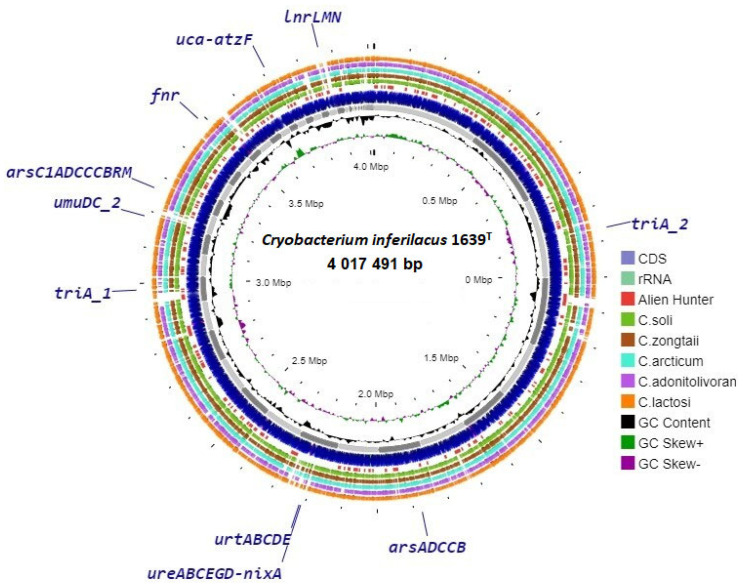
Circular genome map of strain 1639^T^ with BLAST comparison with genomes of closest type strains of *Cryobacterium*. Abbreviations for unique genes: *ure*ABCFGD-*nix*A, urease cluster and transmembrane transporter of nickel ions; *urt*ABCDE, urea transmembrane transporters; *uca-atz*F, urea carboxylase and allophanate hydrolase; *ars*RC1ADCCCBRM, large arsenic resistance operon; *ars*ADCCB, small arsenic resistance operon; *umu*DC, protein of DNA-damage-induced mutagenesis; *lnr*LMN, single multifunctional ABC transporter; *tri*A, melamine deaminase; *fnr*, Crp-Fnr stress regulator.

**Figure 5 microorganisms-13-00990-f005:**
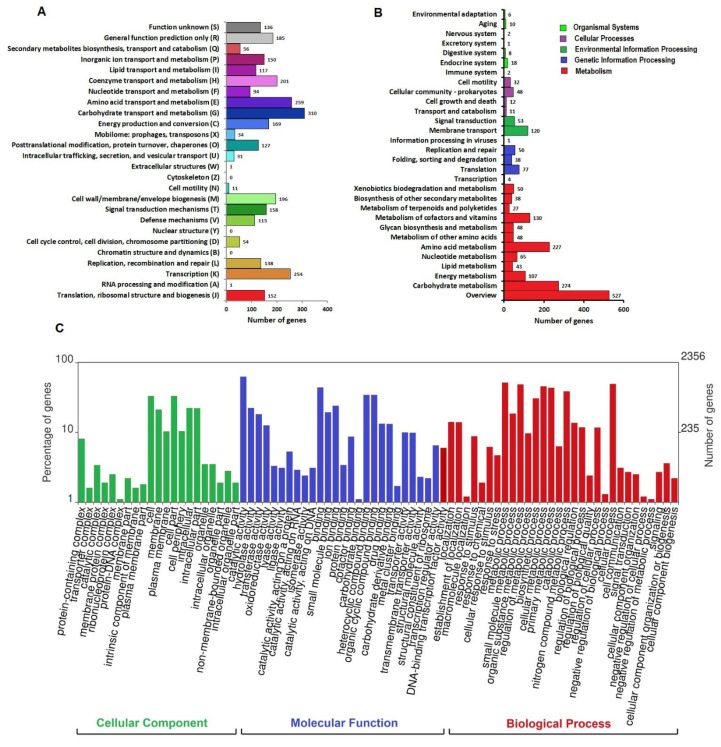
Whole-genome annotation of strain 1639^T^. (**A**) Distribution of Clusters of Orthologous Genes (COG) functional categories in the genome. (**B**) Kyoto Encyclopedia of Genes and Genomes (KEGG) pathway annotation of the genome. Percentage of gene sequences assigned to each subcategory of the five top KEGG Orthology (KO) categories; namely, metabolism (red), genetic information processing (blue), environmental information processing (green), cellular processes (lilac), and cellular processes (light green) were calculated and displayed. (**C**) Gene Ontology (GO) classification of bacterial gene function annotation.

**Figure 6 microorganisms-13-00990-f006:**
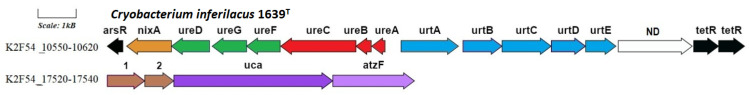
The genes presumably encoding urea assimilation in the genome of strain 1639^T^. Abbreviations: *ure*ABC, subunit of urease; *ure*FGD, urease accessory protein; *nix*A, transmembrane transporter of nickel ions; *ars*R, metalloregulator ArsR family transcription factor; *urt*A, urea ABC transporter, substrate-binding protein UrtA; *urt*B, urea ABC transporter, permease protein UrtB; *urt*C, urea ABC transporter, permease protein UrtC; *urt*D, urea ABC transporter, ATPase protein UrtD; *urt*E, urea ABC transporter, ATPase protein UrtE; ND, putative transmembrane protein; *tet*R, proteins of the TetR family of repressors; *uca*, urea carboxylase; *atz*F, allophanate hydrolase; 1,2, putative urea carboxylase-related aminomethyltransferase.

**Figure 7 microorganisms-13-00990-f007:**
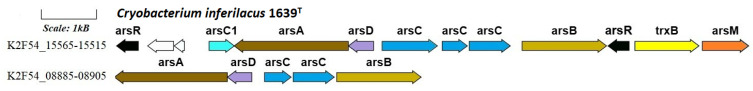
The genes presumably encoding arsenic resistance in the genome of strain 1639^T^. Abbreviations: *ars*R, metalloregulatory protein ArsR/SmtB family transcription factor; *ars*C1, arsenate–mycothiol transferase; *ars*B, arsenite/antimonite:H+ antiporter ArsB; *ars*C, arsenate reductase, thioredoxin-coupled; *ars*D, asenic metallochaperone ArsD; *ars*A, arsenite/antimonite pump-driving ATPase ArsA; *trx*B, thioredoxin reductase; *ars*M, As(III)S-adenosylmethionine methyl transferase.

**Table 1 microorganisms-13-00990-t001:** Comparison of characteristics of strain 1639^T^ and closely related species of the genus *Cryobacterium*.

Characteristic	1	2	3	4	5	6
Colony color	Yellow–green	Light yellow	Smooth yellow	Yellow	Citrine-colored	Lemon yellow
Cell size (µm)	0.3–0.5 × 1.2–1.4	0.8–0.9 × 1.8–2.6	0.3–0.5 × 1.3–2.0	0.2–0.4 × 0.4–2.0	0.7–0.8 × 1.3–1.9	0.7–0.8 × 0.9–1.7
Motility	-	+	-	-	+	+
Source of isolation	Water of Lake Untersee (Antarctica)	Touming Mengke glacier in China	Forest soil	Soil from the bottom of a slope on the shore of Duck Lake, Store Koldewey, north-east Greenland	Glacier samples in China	Melt water of glacier samples in China
Country of origin	Russia	China	China	Germany	China	China
Temperature range (optimum) (°C)	0–25 (10)	4–24 (20)	4–26 (18)	−6–28 (20)	0–26 (14–20)	0–26 (14–20)
pH range for growth (optimum)	4.5–8.7 (6.8–7.8)	6.0–10.0 (7.0)	5.0–11.0 (7.0)	5.0–9.5 (6.5–7.5)	6.0–10.0	6.0–10.0
Substrates:						
mannose	-	+	-	+	+	+
D-lactose	+	+	-	+	+	-
D-arabinose	+	-	+	-	+	+
d-trehalose	+	-	N/D	+	+	+
rhamnose	-	-	+	+	+	N/D
d-galactose	+	+	-	+	+	+
alanine	-	-	N/D	+	+	N/D
proline	-	+	N/D	N/D	+	N/D
formate	-	N/D	N/D	-	+	N/D
lactate	+	-	N/D	-	+	N/D
methanol	+	N/D	N/D	N/D	N/D	N/D
Urea	+	-	-	-	-	-
DNA G+C % content (mol%)	68.03	67.6	68.4	67.8	67.55	67.61
The major fatty acid	i-C_16:0_; ai-C_15:0_; C_16:1_ branched	anteiso-C_15:0_, anteiso A-C_15:1_, iso-_16:0_, anteiso-C_17:0,_ iso-C_17:1_ ω5c	anteiso -C_15:0_, anteiso -C_17:0_	anteiso-_C15:0_, anteiso-C_17:0_ and C_18:0_	anteiso-C_15:0_, iso-C_16:0_, anteiso-C_15:1_, anteiso-C_17:0_	anteiso-C_15:0_, iso-C_16: 0_, anteiso-C_17:0_, iso-C_17:1_ ω5c, anteiso-C_15:1_, anteiso-C_16:0_
Quinone	MK-10	MK-10, MK-11, MK-9	MK-10	MK-10, MK-11	MK-10, with minor amounts of MK-9 and MK-11	MK-10 and MK-9, with minor amounts of MK-11
Cell wall	2, 4-diaminobutyric acid, ornithine, glycine, alanine and glutamic acid	2, 4-diaminobutyric acid	N/D	2,4-Diaminobutyric acid, glycine, alanine and glutamic acid	N/D	N/D
Polar lipids	PG, unidentified phospholipids and glycolipids	DPG, PG, dimannosylglyceride, two unidentified phospholipids and three unidentified lipids	DPG, PG, some unidentified phospholipids and some unidentified polar lipids	DPG, PG	DPG, PG, one unidentified lipid, one unidentified glycolipid	DPG, PG, one unidentified lipid, one unidentified glycolipid

Strains: 1, strain 1639^T^; 2, *C. zongtaii* TMN-42^T^; 3, *C. soli* GCJ02^T^; 4, *C. arcticum* SK1^T^; 5, *C. lactose* Sr59^T^ (=CGMCC 1.9254^T^ = NBRC114035^T^); 6, *C. adonitolivorans* RHLS22-1^T^ (=CGMCC 1.10101^T^ = NBRC 114045^T^). Designations: DPG, diphosphatidylglycerol; PG, phosphatidylglycerol; “+”, positive reaction; “-”, negative reaction; N/D, not determined.

**Table 2 microorganisms-13-00990-t002:** Genes implicated in the adaptation to cold environments in the genome of strain1639^T^.

Functional Classification	Protein Name	Gene Symbol	No. of Genes	RefSeq Locus Tag
DNA replication	RecA protein	*recA*	1	K2F54_08115
	Chromosomal replication initiator protein DnaA	*dnaA*	1	K2F54_11500
DNA transcription	Transcription termination/antitermination protein NusA	*nusA*	1	K2F54_08785
DNA supercoiling relaxation	DNA gyrase subunit A (EC: 5.99.1.3)	*gyrA*	1	K2F54_11500
	DNA-binding protein HU-beta	*hupB*	1	K2F54_11250
Protein folding	Peptidyl-prolyl cis-trans isomerase (EC: 5.2.1.8)	*cypB*	3	K2F54_04485
				K2F54_11555
				K2F54_00570
Protein biosynthesis	Translation initiation factor 1	*infA*	1	K2F54_13655
	Ribosome-binding factor A	*rbfA*	1	K2F54_08800
RNA chaperones	Cold shock protein, CspA family	*cspA*	3	K2F54_09950
				K2F54_00030
				K2F54_09875
Nucleosides and nucleotides	Purine nucleoside phosphorylase (EC 2.4.2.1)	*punA*	1	K2F54_07360
Pyruvate metabolism II	Pyruvate dehydrogenase E1 component (EC 1.2.4.1)	*aceE*	1	K2F54_16525
	Pyruvate dehydrogenase E1 component subunit alpha [EC:1.2.4.1]	*pdhA*	1	K2F54_16525
	Pyruvate dehydrogenase E1 component beta subunit (EC 1.2.4.1)	*pdhB*	1	K2F54_16530
	Pyruvate dehydrogenase E2 component (dihydrolipoyllysine-residue acetyltransferase) [EC:2.3.1.12]	*aceF* *(pdhC)*	2	K2F54_03055
				K2F54_16535
Betaine uptake	Glycine betaine/proline transport system substrate-binding protein	*proX*	1	K2F54_06240
	Glycine betaine/proline transport system permease protein	*proW*	1	K2F54_06245
	Glycine betaine/proline transport system ATP-binding protein [EC:7.6.2.9]	*proV*	1	K2F54_06250
Carotenoid biosynthesis	Phytoene synthase (EC 2.5.1.32)	*crtB*	1	K2F54_02305
	Zeta-carotene-forming phytoene desaturase, EC:1.3.99.29	*crtI*	1	K2F54_02300

## Data Availability

The genome shotgun project of strain 1639T has been deposited at DDBJ/ENA/GenBank under the accession no. JAIEUL000000000, and it is the first version described in this paper.

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
