# Peer review of "Cryobacterium Inferilacus sp. nov., a Pshychrophilic Ureolitic Bacterium From Lake Untersee in Antarctica"

_microorganisms, 2025, doi:10.3390/microorganisms13050990_

Round 1
Reviewer 1 Report
Comments and Suggestions for Authors
The manuscript presents a comprehensive taxonomic and genomic characterization of strain 1639T, a novel psychrophilic bacterium. The authors show that the strain represents a new species within the genus Cryobacterium, supported by phenotypic traits, 16S rRNA analysis, ANI, and functional genomics. The work provides important insights into the fundamental physiology of psychrophilic bacterium.
Comments:
- Have authors described how they experimentally determined the spore formation ability or this is inferred from gene annotation?
- Please include the minimal inhibition concentration for different antibiotics or provide concentrations used in the sensitivity test.
- For the cold-protection genes listed in table 2, I'm not sure about whether some prevalent genes such as recA, dnaA, gyrA, dnaK/J should be there. These genes are commonly found in many bacteria, and authors should analyze if they have particular differences (sequences, structures or domain organization) from the counterparts found in mesophilic bacteria. Their presence in psychrophilic, psychrotolerant, and mesophilic bacteria should be compared.
- Line 407: genomes of five phylogenetically similar strains need to be specified.
Author Response
Dear Review,
Thank you for your careful reading of our article and your fair comments. We have taken into account all your comments and made edits to the text of the article.
Sincerely,
In behalf of all authors
Julia Ju. Berestovskaja
- Have authors described how they experimentally determined the spore formation ability or this is inferred from gene annotation?
The absence of spores in strain 1639T was proven experimentally. Microscopic studies of the morphological features of strain 1639 in light (Olympus CX41, Japan) and transmission electron microscopes (JEM-1400, JEOL, Japan) during its growth at all temperatures studied did not reveal the presence of spores.
Genes responsible for the process of spore formation were also not found in the genome.
2. Please include the minimal inhibition concentration for different antibiotics or provide concentrations used in the sensitivity test.
The concentrations of antibiotics used in the susceptibility testing are included in the text in the section 2. «Materials and Methods», subsection –2.2. «Phenotypic and chemotaxonomic characteristics». (The inserted concentration values are highlighted in red).
3. For the cold-protection genes listed in table 2, I'm not sure about whether some prevalent genes such as recA, dnaA, gyrA, dnaK/J should be there. These genes are commonly found in many bacteria, and authors should analyze if they have particular differences (sequences, structures or domain organization) from the counterparts found in mesophilic bacteria. Their presence in psychrophilic, psychrotolerant, and mesophilic bacteria should be compared.
According to literary data [64,65] the products of the recA, dnaA, gyrA genes are recognized as cold shock proteins, since it has been experimentally proven that the rate of their synthesis under cold shock conditions increases significantly.
Many thanks for pointing out our mistake. Genes dnaK, dnaJ are present in the genome of strain 1639, but do not belong to cold shock genes. We have made corrections to the text and Table 2.
4. Line 407: genomes of five phylogenetically similar strains need to be specified.
In line 407, we inserted the genome numbers of five strains of the closely related to strain 1639T.
(Complete genome sequence numbers are inserted in line 407 in red.)
Reviewer 2 Report
Comments and Suggestions for Authors
Yu et al, Present an interesting study in which they describe morphology, physiology, and chemotaxonomic features of the strain 1639T isolated from water sample of subglacial low-temperature, in order to support its affiliation to the genus Cryobacterium , the study is really well conducted and provide relevant information about taxonomic classification of Cryobacterium genus . however there are some concerns that must are addressed :
- In the introduction section it would be appropriate to provide relevant information about the bacterial ecology of this new specie or / and it genus, its potential environmental role or epidemiological or clinical significance, in order rise the interesting about the study of this strain or related ones.
- in the discussion section please on the subheading Genome functional characterisation, please provide a more deeper discussion about relevant important information with high biotechnology potential such as : Xenobiotics degradation and metabolism
- finally , It is evident that the general objective of the study is the classification of this new bacterial species Cryobacterium inferilacus strain 1639T
and the singular observation of being the first species belonging to Cryobacterium genus to use ammonium nitrogen as a nitrogen source. please provide potential use and effects of this bacteria in the nitrogen cycle
Author Response
Dear reviewer, thank you for your interest in our work and your comments.
The text of the article has been supplemented with answers to your comments. They are highlighted in red.
On behalf of all the authors,
Ju.Ju. Berestovskaja
1.In the Introduction section, we presented only published information about selected species of the genus Cryobacterium, which were described taxonomically by the authors. The published articles did not contain any information about the epidemiological or clinical significance of these bacteria. Therefore, it is not possible for us to provide such information.
2. According to the BlastCOALA portal, no genes of complete pathways for xenobiotic degradation were found. However, in section 3.10. Arsenic resistance, a suggestion about the possible potential of strain 1639T in arsenic bioremediation was added.
3. Strain 1639 uses ammonium nitrogen, which is formed in the nitrogen cycle as a result of the work of the ammonifying microflora, as a nitrogen source.
Reviewer 3 Report
Comments and Suggestions for Authors
This work presents a new bacterial species. It is worth highlighting that some of the most relevant aspects of the biology of an organism were analyzed to ensure its presentation to the scientific community with the necessary rigor.
An adequate description of the culture media used to grow the microorganism was provided, its biochemical composition was adequately described, and a rigorous analysis of its genome was performed to utilize the most relevant databases for its taxonomic classification.
An interesting element is a hypothesis that explains why this species is capable of utilizing ammonium nitrogen as a carbon source. This is undoubtedly an interesting contribution from the perspective of microbial ecology, which also allows us to anticipate the ecosystem conditions under which other species belonging to this genus might be found.
In my opinion, this work, although a descriptive study, is what this type of work should necessarily include, and therefore, in my opinion, it could be published.
Author Response
Dear Review,
We thank you for your careful reading of the article and your high appreciation of our study of a new organism of the genus Cryobacterium.
Sincerely,
In behalf of all authors
Julia Ju. Berestovskaja